# Elevated psychological distress in undergraduate and graduate entry students entering first year medical school

**Sean R. Atkinson**⬦*

School of Rural Health, Monash University, Churchill, Australia

* sean.atkinson@monash.edu

## Abstract

### Background

Psychological distress in medical students is a global issue and poses a risk to their health, academic performance, and ability to care for patients as clinicians. There has been limited research on psychological distress levels in students prior to starting medicine and no direct comparison between undergraduate and graduate-entry students.

### Methods

Psychological distress was assessed using the 21-item Depression Anxiety and Stress Scale in 168 undergraduate-entry and 84 graduate-entry medical students at two separated campuses of the same university in the orientation week prior to starting classes. Mean scores and severity proportions were compared between the two cohorts of students. Demographic data was also collected and compared to distress scores using subgroup analysis.

### Results

The response rate for the study was 60.9%. The majority of undergraduate and graduate-entry medical students were within the normal limits for depression (67.2% versus 70.2%, p = 0.63), anxiety (56.5% versus 44.0%, p = 0.06), and stress scores (74.4% versus 64.2%, p = 0.10). There was no significant difference between severity groups except for severe stress (2.3% versus 9.5%, p = 0.01). The mean scores of the clinically distressed groups indicated moderate levels of depression, moderate anxiety, and moderate stress scores. There were no significant differences between undergraduate or graduate-entry students for depressive ($\bar{x}$ = 17.02 versus 15.76, p = 0.43), anxiety ($\bar{x}$ = 14.22 versus 13.28, p = 0.39), and stress scores ($\bar{x}$ = 20.83 versus 22.46, p = 0.24). Female gender and self-believed financial concerns were found be associated with higher levels stress in graduate entry students.

### Conclusions

The majority of medical students enter medical school with normal levels of psychological distress. However, a large number of undergraduate and graduate-entry medical students

**Data Availability Statement:** All relevant data are within the paper and its Supporting Information files.

**Funding:** The author received no specific funding for this research.

**Competing interests:** The authors have declared that no competing interests exist.

**Abbreviations:** DASS-21, Depression, anxiety, and stress scale; ANOVA, analysis of variance.

have significant levels of depressive, anxiety, and stress levels, without a significant difference between undergraduate or graduate-entry students. There are several limitation of this study but the results suggest that education and intervention may be required to support students from the earliest weeks of medical school.

## Introduction

Medical students have higher rates of stress and mental health problems than their matched peers and it appears to be a global phenomenon [1–3]. Research suggests that at least one third and possibly up to one half of medical students show some form of psychological distress during medical school [3]. Risk factors put forth as contributing to distress, include perfectionist and neurotic traits, academic workload, sleep issues, exposure illness and death of patients, culture, and parenting styles [3–5].

Identifying students in psychological distress is vital as it has been associated with poor academic performance, decreased empathy, medical errors, depression, anxiety, and suicidality [6–9]. Research has focussed on students already at medical school but there is less research on students prior to commencement and scarce data comparing undergraduate to graduate-entry students [10]. This information would establish whether students start the course in psychological distress or whether it becomes the result of undertaking the course.

In Australia there are two entry points into medical school. Undergraduate students enter the course from high school and graduate-entry students who enter the course after obtaining an undergraduate degree [11]. The coping styles differ between these cohorts and so it is important to consider both groups when performing research [12].

### Aim

The aim of this research was to determine the baseline level of psychological distress in graduate and undergraduate students prior to commencing their first year of medical school.

## Method

### Participants

Participants were invited to join in the study from two cohorts of first year medical students from one Australian university in January 2019. Each cohort attended a different campus with undergraduate-entry students from the urban campus and graduate-entry students from the rural campus. The undergraduate cohort is larger than the graduate-entry group based upon student intake each year.

### Sample size

A sample size calculation was performed it was determined that the study needed 200 completed surveys to achieve a 95% confidence level with a presumed population proportion of 50%.

### Inclusion and exclusion criteria

To be included participants had to be a first year medical student attending the university at either campus. Students were excluded if they had previously been a first year medical student at another university or were repeating the year.

## Data collection

The questionnaire was emailed out to all first year students attending both campuses at the start of the orientation week and available for that week only. All data was obtained prior to the starting of classes which began in the following week to avoid confounding. All participation was voluntary and students could only participate if they were above 18 years of age. Informed consent was obtained from all participating students prior to data collection.

## Ethics

Ethics approval was obtained from the Monash University Ethics Committee in January 2019. Approval number 2018-17900-26467

## Measures

All participants completed demographic information and the 21-item Depression, Anxiety, and Stress Scale (DASS-21) which is a measure of psychological distress which has been used in medical student populations, with good reliability and validity for evaluation of psychological distress [13].

Participant completed the 21 Likert style questions of the DASS-21 and the scores and severity levels were calculated following the method set out by DASS-21 questionnaire manual [14]. A clinically significant result was determined as any score above the normal range in each category of depression, anxiety, stress, while severity levels were then applied to any abnormal scores (Table 1).

## Statistical analysis

Proportional differences in demographic data between undergraduate-entry and graduate-entry medical students were compared using chi-square testing with significance set at <0.05.

Mean scores on the DASS-21 for undergraduate and graduate-entry students were compared in each of the three variables of depression, anxiety, stress using a two-tailed t-test with a significance testing set at $p < 0.05$.

DASS-21 scores were then categorised as 'normal', 'mild, 'moderate', 'severe', and 'extremely severe' as per the DASS-21 manual [14]. These groups were compared with Fisher's exact test with significance with a $p < 0.05$.

Medical student scores for depression, anxiety, and stress were also compared with age, gender, rurality experience of 5 years or more, nationality, relocation for medical school, and part-time work, using one-way analysis of variance with Tukey's post-hoc testing for post hoc analysis of significant results.

Data analysis was performed using Microsoft Excel (Microsoft Office Professional Plus 2016) and MedCalc 19.3.1 for Windows.

**Table 1. Depression, anxiety, and stress scale (DASS-21) scoring.**

| Severity | Depression | Anxiety | Stress |
|---|---|---|---|
| Normal | 0–9 | 0–7 | 0–14 |
| Mild | 10–13 | 8–9 | 15–18 |
| Moderate | 14–20 | 10–14 | 19–25 |
| Severe | 21–27 | 15–19 | 26–33 |
| Extremely severe | 28+ | 20+ | 34+ |

**Table 2. Undergraduate and graduate-entry student comparative demographic data with chi-square analysis for proportional differences.**

|  |  | Graduate-entry | Undergraduate-entry | Chi-square | p-value |
|---|---|---|---|---|---|
| **Total participants** |  | 84 | 168 |  |  |
| **Gender** | **Male** | 71 (42.2%) | 30 (35.7%) | 0.37 | 0.54 |
|  | **Female** | 97 (57.8%) | 54 (64.3%) | 0.61 | 0.44 |
| **Average age (years)** |  | 22.4 | 18.5 |  |  |
| **Nationality** | **Australian** | 56 (66.6%) | 128 (75%) | 1.96 | 0.16 |
|  | **Permanent resident** | 2 (2.4%) | 5 (3%) | 0.07 | 0.79 |
|  | **International** | 26 (31%) | 37 (22%) | 0.88 | 0.35 |
| **Rurality experience** | **Yes** | 29 (34.5%) | 28 (16.6%) | 10.23 | <0.01 |
|  | **No** | 55 (65.5%) | 140 (83.3%) | 10.09 | <0.01 |
| **Current part time work** | **Yes** | 24 (28.6%) | 106 (63.1%) | 26.58 | <0.01 |
|  | **No** | 60 (71.4%) | (36.9%) | 26.58 | <0.01 |
| **Financial concern** | **Yes** | 61 (72.6%) | 61 (36.3%) | 29.43 | <0.01 |
|  | **No** | 23 (27.4%) | 107 (63.7%) | 29.43 | <0.01 |
| **Relocation** | **Yes** | 84 (100%) | 76 (45.2%) | 72.24 | <0.01 |
|  | **No** | 0 (0%) | 92 (54.8%) | 72.24 | <0.01 |

## Results

### Participants

Invitations to participate were sent to 414 first year medical students of which 252 students completed the questionnaire in its entirety and were included in the analysis (60.9%). There was no significant differences in gender or proportion of Australian and international students between the cohorts. However, there were significant differences for rurality experience, part-time work, financial concerns, and relocating for medical school (Table 2).

### Psychological distress based upon DASS-21 categories of severity

The students who experienced normal DASS-21 scores were in the majority for both cohorts. There was a significant difference for the severe category of stress with graduate-entry students having more stressed students (9.5% versus 2.3%, p = 0.02). However, there was no other significant differences identified (Table 3). In summary, it can be inferred that around 1 in 3 students has clinically significant depressive and/or stress symptoms, and almost 1 in 2 has clinically significant anxiety symptoms.

### Psychological distress of the clinically significant compared to those in the normal groups

The means of the clinically distressed groups indicated moderate levels of depression, moderate anxiety, and moderate stress scores (Table 4). All results were significantly different and clinically different to the not-distressed 'normal' groups.

**Table 3. DASS-21 scores of undergraduate and graduate-entry students in severity groups with significance determined by Fisher's exact test.**

| % | Depression | | | Anxiety | | | Stress | | |
|---|---|---|---|---|---|---|---|---|---|
|  | Graduate | Undergraduate | p-value | Graduate | Undergraduate | p-value | Graduate | Undergraduate | p-value |
| **Normal** | 70.2 | 67.2 | 0.67 | 44.0 | 56.5 | 0.08 | 64.2 | 74.4 | 0.11 |
| **Mild** | 9.5 | 10.7 | 0.83 | 13.1 | 10.1 | 0.53 | 14.2 | 11.3 | 0.54 |
| **Moderate** | 15.4 | 14.2 | 0.42 | 25.0 | 17.8 | 0.19 | 8.3 | 10.7 | 0.66 |
| **Severe** | 2.3 | 4.7 | 0.50 | 10.7 | 6.5 | 0.32 | 9.5 | 2.3 | 0.02 |
| **Extreme** | 2.3 | 2.9 | 0.99 | 7.1 | 8.9 | 0.81 | 3.5 | 1.1 | 0.33 |

**Table 4.  Two way t-test analysis of DASS-21 means for psychologically distressed undergraduate and graduate-entry students compared to means of those within the normal range.**

|  | Normal Range | Mean | STD-DEV | Normal | STD-DEV | p-value |
|---|---|---|---|---|---|---|
|  | D (0–9) | 15.76 | 5.49 | 3.19 | 2.78 | <0.01 |
| Graduate | A (0–7) | 13.28 | 5.70 | 3.51 | 2.02 | <0.01 |
|  | S (0–14) | 22.46 | 6.07 | 8.70 | 3.86 | <0.01 |
|  | D (0–9) | 17.09 | 7.05 | 3.68 | 2.47 | <0.01 |
| Undergraduate | A (0–7) | 14.22 | 5.94 | 3.20 | 2.17 | <0.01 |
|  | S (0–14) | 20.83 | 5.56 | 7.55 | 4.37 | <0.01 |

There were no significant differences between undergraduate or graduate-entry students for depressive ($\bar{x}$ = 17.02 versus 15.76, p = 0.43), anxiety ($\bar{x}$ = 14.22 versus 13.28, p = 0.39), and stress scores ($\bar{x}$ = 20.83 versus 22.46, p = 0.24).

## Subgroup analysis

**Gender.**   One-way ANOVA showed a significant difference when stress scores were compared to gender in graduate and undergraduate students (Table 5). However, there was no significance to depression and anxiety symptoms. Post hoc analysis indicated that female graduate-entry students ($\bar{x}$ = 7.31, SD = 4.1) scored significantly higher on stress scores than male undergraduate students ($\bar{x}$ = 4.85, SD = 3.37). There was no other significantly associations.

**Financials.**   One-way ANOVA showed a significant difference when stress scoring was compared to self-believed financial concerns (Table 5). Again, there was no significant association to depression and anxiety symptoms. Post hoc analysis indicated that graduate-entry students ($\bar{x}$ = 6.90, SD = 4.46) scored significantly higher on stress scores than undergraduate students without financial concerns ($\bar{x}$ = 5.07, SD = 3.64). There was no other significantly associations.

**Table 5.  ANOVA of multiple variables for undergraduate and graduate-entry medical students.**

| Variable | DASS-21 | df | F value | p value |
|---|---|---|---|---|
| Gender | Depression | 3 | 2.27 | 0.08 |
|  | Anxiety | 3 | 2.52 | 0.06 |
|  | Stress | 3 | 4.45 | 0.005 |
| Nationality | Depression | 2 | 0.19 | 0.89 |
|  | Anxiety | 2 | 0.88 | 0.42 |
|  | Stress | 2 | 1.14 | 0.32 |
| Financial concern | Depression | 3 | 1.26 | 0.29 |
|  | Anxiety | 3 | 1.41 | 0.24 |
|  | Stress | 3 | 3.36 | 0.02 |
| Rurality experience | Depression | 3 | 0.45 | 0.71 |
|  | Anxiety | 3 | 1.79 | 0.14 |
|  | Stress | 3 | 2.16 | 0.09 |
| Relocation | Depression | 6 | 0.30 | 0.94 |
|  | Anxiety | 6 | 0.73 | 0.63 |
|  | Stress | 6 | 1.64 | 0.14 |
| Current part-time employment | Depression | 3 | 0.92 | 0.43 |
|  | Anxiety | 3 | 1.36 | 0.25 |
|  | Stress | 3 | 2.45 | 0.06 |

**Other demographics.**   There was no significant differences between the groups in any of the variables for relocation, nationality, previous employment, or rurality experience of at least 5 years (Table 5).

## Discussion

The majority of the students scored within the normal range for depressive, anxious, and stress symptoms. One third of students showed significant stress and depressive symptoms while a half of students were anxious, with little differences between the two cohorts. The proportions are similar with other results found in Australia. However, obtaining a prevalence is difficult to elucidate due to the use of a variety of psychological measurements of distress, mixed years of study, and type of entry [1, 15–19]. Similarly, two global meta-analyses of mixed year levels for anxiety and depression found similar proportions to this study [3, 8]. While comparisons are limited and caution used in making bold inferences due to study differences, the result do suggest a possible commonality of experience that seems to transcend culture and borders.

This study reveals that graduate-entry students, who have previous experience with a university course, do not have a reduced burden of psychological distress as compared with their younger, school-leaving colleagues. This has also been found by others in Australia and internationally in mixed year levels and here has yet to be a clear answer as to the absence of differences between these two cohorts [1, 12]. Clarity of this issue has also eluded this author. However, it could be hypothesised that psychological distress is dependent on new or unaccustomed activities and so age in itself is not relevant. Further, graduate and undergraduate cohorts are not often significantly dissimilar in age as opposed to generational differences in maturity. Finally, perhaps medical schools inadvertently select specific personality traits which may predispose to psychological distress. Further research should be conducted to help fully clarify this unexpected outcome.

Importantly, this study shows that students are entering medical school in distress rather than it developing as a consequence of exposure to medical school and that entry-type is not associated with a difference in the presence of any marker of distress or its severity. This is an important shift from what others have found earlier, for both presence of psychological distress and its severity [10, 20].

Further supporting this theory is a study of pre-medical undergraduate students that showed psychological distress and burnout at higher rates than other undergraduates [21]. There is scant evidence for psychological distress in high school leavers entering medicine. However, a recent study showed that high school students interested in studying medicine experienced stress levels equal to early-years medical students [22]. Taken together, it may be suggested that times have changed and graduate and undergraduate-entry students could now share commonality in early experiences that predispose them to higher rates of psychological distress. However, this is yet to be fully elucidated.

Psychologically distressed medical students often show maladaptive perfectionism, cognitive distortions, imposter syndrome, and negative feelings such as shame and embarrassment [23]. These traits are learned behaviours and would likely have been deeply rooted prior to medical school [24]. At least for depressive symptoms that personal factors such as personality traits and relationships may contribute more to the maintenance of symptoms than other factors such as medical school [25]. Overall, it may be more appropriate to consider medicine as a new stressor and not the main driver for student distress.

There was limited differences between the students in subgroup analysis. Of note, international students were not more psychologically distressed on any of the scales compared to Australian or permanent residents. This was an interesting finding given the challenges faced by

international students studying in a foreign country [26]. Similarly, rural background students were also not more psychologically distressed than their urban counterparts. This is interesting as rural origin has been linked as a possible risk factor for burn out in later years and higher rates of self-reported stress [27, 28]. It is possible that these groups may be more adaptable to change than previously considered and there is some evidence for this [29].

Financial issues can be a significant issue to medical students [30]. Graduate-entry students with self-believed financial concerns showed significant higher levels of stress. This seems logical as they are beginning their second degree and will therefore be facing a more significant financial debt. Furthermore, all graduate students in this study were required to relocate and many were unemployed which may increase their financial concern. This would make sense as there was no differences in stress whether students simply relocated or not.

In this study there was a significant difference between graduate-entry female students and their male undergraduate counterparts in stress scores. There was no other associations. Previous research has been inconsistent when gender is compared to depression, anxiety, and stress which suggests that gender likely has no significant role as a risk factor for psychological distress [1, 3, 8, 31, 32].

If students are entering the course with distress it is important to consider whether initial screening of candidates for psychological distress is appropriate [33]. It has been suggested that the addition of psychological screening may allow for selection of more resilient medical students [34–36]. Psychological screening is used in other high stress jobs such as police, military, and airline pilots [37, 38]. However, there is also the risk of prejudice against those with stable, or previous mental health conditions by suggesting they are incapable of undertaking a medical career, of which there is limited evidence [33]. Finally, while not researched, there is the possibility for cheating if using standardised psychological tests to obtain a desired result during the screening process due to the competitive nature of entering medicine.

It would seem appropriate that early interventions are needed if students are entering medical school with psychological distress. To date many of the interventions involving pre-clinical medical students have yielded mixed results [39–41]. Furthermore, a recent review suggested short term but not long term effectiveness of many interventions for medical students [42].

A general consensus statement from Australia and New Zealand suggested a broad and integrated approach was needed to help psychologically distressed students [33]. Their report suggested among other things that medical schools can assist to encourage self-awareness through educational sessions, de-stigmatise distress, and encourage help-seeking to appropriate healthcare providers. This study suggests from the results that engagement needs to occur in the earliest weeks. This approach may allow staff to perform their academic role without becoming surrogates for healthcare professionals. Simultaneously, this may increase student mental health literacy and health seeking behaviour [43]. Furthermore, this strategy may also reduce the stress on staff, who themselves are at risk of mental health problems and burn out [44, 45]. However, there are some barriers to consider such as student's perceived risk to progression, cost, distance and leave for appointments, and academics comfort with pastoral care that also need to be addressed [46–48]. Finally, it is important that medical schools discourage an environment where psychological distress can fester through an evolution in curriculum design that promotes collaboration not competition, improvement not perfection, collegiality not hierarchy, and construction not criticism [35].

## Limitations

This research looked specifically at students about to start medical school and does not provide any information on how student psychological distress changes over time. It is possible that

starting a new course may have increased their stress levels and it could decline with time. The graduate cohort was much smaller than the undergraduate cohort and this may have under- or overestimated an effect size. Importantly, while a reasonable amount of students participated in this study it would have been more powerful with a higher participation rate. Furthermore, with almost forty percent of students not participating in the study there is a risk of selection bias which could affect the generalisability of the data into real-world effects. However, while the prevalence of how many students could be psychologically distressed may change with increased participation in the study it is important to note that the results of this study appear to align with what has been previously found.

There is a risk that unidentified factors may be able to explain the psychological distress observed in the students and lack of differences between the cohorts. The study attempted to control for confounders in the design by analysing the two cohorts independently, examining the data of the normal and abnormal results separately to avoid dilution of the severity of scores, and looked at several possible demographic risk factors. It is likely that there are other factors present that may have contributed to the results observed. Possible factors may have included a student's past or current mental health diagnoses, whether medical students with psychological distress participate more or less in research, and honesty in reporting based upon their feelings of stigma, may have contributed to the results [49]. However, this was not examined in this study and may be beneficial in future research efforts.

## Further research

Further research needs to be aimed primarily at the underlying psychological causes of distress. It is important that appropriate evidence based strategies are implemented to help identify students in need and at risk of psychological distress. Once students are identifiable then it is important for medical schools to find effective methods to connect them with an appropriate healthcare professional and to understand the barriers and solutions to access to care. It is also important to study, identify, and evolve curricula to meet the psychological needs of students. Finally, it is medical school staff who are the vanguard of this issue and so it is important to see how student distress affects their mental health.

## Conclusion

This study of undergraduate and graduate-entry first year medical students showed that at least one third of students showed some form of psychological distress prior to commencement. There was no significant differences in depression, anxiety or stress mean scores or severity between the cohorts despite age and attending different campuses. Within the limitations of this study, this study suggests that all students may need appropriate education, identification, and intervention from the earliest weeks.

## Supporting information

**S1 Data.**
(XLSX)

## Acknowledgments

My thanks go to Dr. David Reser for his invaluable advice on improving my academic writing for this paper

## Author Contributions

**Conceptualization:** Sean R. Atkinson.

**Formal analysis:** Sean R. Atkinson.

**Writing – original draft:** Sean R. Atkinson.

**Writing – review & editing:** Sean R. Atkinson.

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
