## [Decision Letter · Decision Letter 0]

19 May 2020

PONE-D-20-04775

Distress in undergraduate and graduate entry medical students at the start of first year medical school is clinically significant

PLOS ONE

Dear Dr. Atkinson,

Thank you for submitting your manuscript to PLOS ONE. After careful consideration, we feel that it has merit but does not fully meet PLOS ONE’s publication criteria as it currently stands. Therefore, we invite you to submit a revised version of the manuscript that addresses the points raised during the review process.

The revisions suggested for the methods section are important for further evaluation of the manuscript. In addition, the limitations need to be explicitly discussed in the interpretation of the data. 

We would appreciate receiving your revised manuscript by June 19, 2020. To enhance the reproducibility of your results, we recommend that if applicable you deposit your laboratory protocols in protocols.io, where a protocol can be assigned its own identifier (DOI) such that it can be cited independently in the future. For instructions see: http://journals.plos.org/plosone/s/submission-guidelines#loc-laboratory-protocols

We look forward to receiving your revised manuscript.

Kind regards,

Markos Tesfaye, M.D., Ph.D

Academic Editor

PLOS ONE

Additional Editor Comments:

The manuscript reports findings from a cross-sectional study of depressive, anxiety and stress among medical students using structured self-administered questionnaire. The findings use chi-square, t-test, and ANOVA to identify associations between exposure and outcome variables. While the area of research is important for people working in the field of mental health, there are important issues missing from the report that limit its scientific value. The authors need to address the following issues to improve the report.

- The title of the manuscript appears to imply that the authors main aim was to test a distress measured by some tool was clinically significant or not. However, the authors have not conducted a gold-standard assessment for clinical significance. I suggest that the title is revised to fit the objectives of the study.

- Please use the term "psychological distress" or "mental distress" rather than "distress" - which may also include respiratory distress, etc.

- The methods section lacks important details such as sample size calculation, if any sampling technique has been used, inclusion and exclusion criteria. In addition, the description given regarding the measurement tool is scanty and advises readers to refer to the manual without providing reference. The details of the scoring, cut-offs, and reference to the validity of the tool need to be given. As the authors have written about "significant distress" in the subsequent sections, what constitutes "clinically significant distress" should be clearly defined in the methods.

- The use of chi-square tests for data with a few subjects in the categories is not recommended. Perhaps the use of Fisher's exact test could address the issue for the results reported in Table 2.

- The appropriate statistical methods for addressing confounding has not been used. Furthermore, the limitations which might arise from the potential confounder variables is not addressed as a limitation.

- Nearly 40% of those who are invited have either declined or did not have data included. The potential selection bias which might affect the reported prevalence is not discussed. In fact, there is evidence that people with poorer mental health might be more willing to participate - which would bias the prevalence estimates.

- The discussion of the prevalence needs to be more detailed. In particular, comparison with pooled prevalence from meta-analyses may have limitation because of the different tools used and cultural settings across the globe. I suggest some discussion of prevalence with studies from the region , if possible using the same criteria.

- The limitations mentioned above need to be addressed.

- The language needs to be revised, the use of "statistically significantly" and other typos need to be revised.

2. Please provide additional details regarding participant consent. In the ethics statement in the Methods and online submission information, please ensure that you have specified whether consent was informed.

3. Please include a copy of Table 5 which you refer to in your text on page 7.

5. Your ethics statement must appear in the Methods section of your manuscript. If your ethics statement is written in any section besides the Methods, please move it to the Methods section and delete it from any other section. Please also ensure that your ethics statement is included in your manuscript, as the ethics section of your online submission will not be published alongside your manuscript.

Reviewers' comments:

Reviewer's Responses to Questions

**Comments to the Author**

1. Is the manuscript technically sound, and do the data support the conclusions?

Reviewer #1: Partly

Reviewer #2: Yes

Reviewer #3: Yes

2. Has the statistical analysis been performed appropriately and rigorously? 

Reviewer #1: Yes

Reviewer #2: Yes

Reviewer #3: Yes

3. Have the authors made all data underlying the findings in their manuscript fully available?

Reviewer #1: Yes

Reviewer #2: Yes

Reviewer #3: Yes

4. Is the manuscript presented in an intelligible fashion and written in standard English?

Reviewer #1: Yes

Reviewer #2: Yes

Reviewer #3: Yes

5. Review Comments to the Author

Reviewer #1: The manuscript covers a relevant topic with an original point of view, and it could be of interest for Plos One’s readers. Method and data analysis are very interesting, however I would suggest some minor changes:

- At page 4, in the section “measures”, please give some psychometric details on the tool, in order that the reader can understand the validity of the measure; is there a specific cut-off for the scale that suggest you to talk about “clinically significant symptoms”? If yes, please describe it better, if not please rephrase the sentences in order to avoid to give judges not based on clinical cut-offs.

- At pag 5, in the Participants’ section, you wrote that 252 students entirely completed the questionnaire. In order to make it cleare for readers, the percentage should be calculated on the final participants enrolled, so it won’t be 60.9%.

- At pag. 6, when talking about stress in the two different sub-samples, in the text you talk of a p=0.001, while in the table you report p=0.01. Please uniform the values.

- At pag. 6, when discussing Table 2, you write “it can be inferred that around 1 in 3 students has clinically significant depressive and stress symptoms”. I would suggest to change in “depressive and/or stress symptoms”, since we do not know if students with depressive and stress symptoms are the same (this aspect should be replicated also in the discussion section).

- At pag. 7, both in the gender and in the financial sections, you referred to Table 5, while probably you should write “Table 4”.

- At pag. 10, this sentence is not clear: “This makes sense as medicine is their second degree ad so they have been longer in financial debt.”. Please, rephrase it in order to make it clearer than before.

- At pag. 12, please include all the other abbreviations that you use.

Moreover, specific limitations of the study should be considered when discussing conclusions, which should be more cautious and less categorical.

Reviewer #2: The manuscript is interesting and well written, I only suggest some points to be addressed:

- Authors should specify the study period and they way used for the sample size calculation: convenience sample?

- Authors should cite the software used for the research

Reviewer #3: The study is very interesting and analyzes the problem of medical students' distress from different points of view.

On of the most intersting findings is that graduate-entry students, who have previous experience with a university course, do not have a reduced burden of distress as compared with their younger, school-leaving colleagues.

However no explanations are suggested, and although the author explicitly says that further research needs to be aimed primarily at the underlying psychological causes of distress, it would be interesting to provide some hypotheses.

6. PLOS authors have the option to publish the peer review history of their article (what does this mean?). If published, this will include your full peer review and any attached files.

Reviewer #1: No

Reviewer #2: No

Reviewer #3: No

---

## [Author Response · Author response to Decision Letter 0]

11 Jun 2020

I have responded in the attached letter

---

## [Editor Report · Decision Letter 1]

29 Jun 2020

PONE-D-20-04775R1

Elevated psychological distress in undergraduate and graduate entry students entering first year medical school

PLOS ONE

Dear Dr. Atkinson,

Thank you for submitting your manuscript to PLOS ONE. After careful consideration, we feel that it has merit but does not fully meet PLOS ONE’s publication criteria as it currently stands. Therefore, we invite you to submit a revised version of the manuscript that addresses the points raised during the review process.

We look forward to receiving your revised manuscript.

Kind regards,

Markos Tesfaye, M.D., Ph.D

Academic Editor

PLOS ONE

Additional Editor Comments (if provided):

The revised manuscript appears to have addressed the comments provided previously. However, there are still minor but important corrections needed given below:

1. The reference no. (14) cited on pages 5 and 6 to refer to the "manual" is not actually the manual. It should be changed to reference no. 13. Please also use the standard and complete citation details for reference no 13 in the list of references.

2. "Other demographics" on page 9 refers to Table 4 while the results are in Table 5. Please correct.

3. The section on sample size calculations is better placed immediately after the "Participants" section and before "inclusion and exclusion" criteria.

4. It is customary to write the title of the tables on the top of the tables rather than below.

---

## [Author Response · Author response to Decision Letter 1]

9 Jul 2020

I have corrected the minor changes as per the editor. There was no reviewer questions on this revision. I have added supporting documents for the raw data.

---

## [Editor Report · Decision Letter 2]

20 Jul 2020

Elevated psychological distress in undergraduate and graduate entry students entering first year medical school

PONE-D-20-04775R2

Dear Dr. Atkinson,

We’re pleased to inform you that your manuscript has been judged scientifically suitable for publication and will be formally accepted for publication once it meets all outstanding technical requirements.

Kind regards,

Markos Tesfaye, M.D., Ph.D

Academic Editor

PLOS ONE

---

## [Editor Report · Acceptance letter]

27 Jul 2020

PONE-D-20-04775R2 

Elevated psychological distress in undergraduate and graduate entry students entering first year medical school 

Dear Dr. Atkinson:

I'm pleased to inform you that your manuscript has been deemed suitable for publication in PLOS ONE. Congratulations! Your manuscript is now with our production department. 

Kind regards, 

on behalf of

Prof. Markos Tesfaye 

Academic Editor

PLOS ONE